# Evaluating the African food supply against the nutrient intake goals set for preventing diet-related non-communicable diseases: 1990 to 2017 trend analysis

Samson Gebremedhin[1]*, Tilahun Bekele[2]

**1** School of Public Health, Addis Ababa University, Addis Ababa, Ethiopia, **2** Center for Food Science and Nutrition, Addis Ababa University, Addis Ababa, Ethiopia

* samsongmgs@yahoo.com

## Abstract

### Background

Population intake goals intended to prevent diet-related non-communicable diseases (NCDs) have been defined for multiple nutrients. Yet, little is known whether the existing food supply in Africa is in conformity with these goals or not. We evaluated the African food balances against the recommendations for macronutrients, free sugars, types of fatty acids, cholesterol and fruits and vegetables over 1990 to 2017, and provided regional, sub-regional and country-level estimates.

### Methods

The per capita supply of 95 food commodities for 45 African countries (1990–2017) was accessed from the FAOSTAT database and converted into calories, carbohydrate, fat, protein, free sugars, cholesterol, saturated (SFA), monounsaturated (MUFA), and polyunsaturated (PUFA) fatty acids contents using the Food Data Central database. The supply of fruits and vegetables was also computed.

### Results

In Africa the energy supply increased by 16.6% from 2,685 in 1990 to 3,132 kcal/person/day in 2017. However, the energy contribution of carbohydrate, fat and protein remained constant and almost within acceptable range around 73, 10 and 9%, respectively. In 2017, calories from fats surpassed the 20% limit in upper-middle- or high-income and Southern Africa countries. Energy from SFA remained within range (<10%) but that of PUFA was below the minimum desirable level of 6% in 28 countries. Over the period, energy from free sugars remained constant around 7% but the figure exceeded the limit of 10% in upper-middle- or high-income countries (14.7%) and in Southern (14.8%) and Northern (10.5%) sub-regions. Between 1990 and 2017 the availability of dietary cholesterol per person surged by 14% but was below the upper limit of 300 mg/day. The supply of fruits and vegetables increased by

**Data Availability Statement:** Data are freely available from the FAOSTAT database. Please use the following links to access the data: http://www.

fao.org/faostat/en/#data/FBSH http://www.fao.org/faostat/en/#data/FBS.

**Funding:** The authors received no specific funding for this work.

**Competing interests:** The authors have declared that no competing interests exist.

27.5% from 279 to 356 g/capita/day; yet, with the exception of Northern Africa, the figure remained below the target of 400 g/capita/day in all sub-regions.

## Conclusion

According to this population level data, in Africa most population intake goals are within acceptable range. Yet, the supply of fruits and vegetables and PUFAs are suboptimal and the increasing energy contributions of free sugars and fats are emerging concerns in specific sub-regions.

## Background

Every year 41 million deaths, equivalent to 71% of the total deaths, occur globally due to non-communicable diseases (NCDs) [1]. Between 1990 and 2017, the number of avertable NCD-related deaths increased by nearly 50% [2]. Specifically, cardiovascular diseases, cancers, chronic respiratory diseases and diabetes contribute to more than 80% of the total NCD-related deaths [1]. With the existing trends, the Sustainable Development Goals (SDGs) target to reduce pre-mature mortality from NCDs by one-third is unlikely to be realized [3].

NCDs once considered as diseases of affluence are now disproportionately affecting low- and middle-income countries (LMIC) [4]. The risk of death from NCDs is now higher in LMIC than high-income countries and more than 80% of the global NCD-related deaths now occur in LMIC [2, 4]. According to the recent estimate of the World Health Organization (WHO), in Africa 41% of the total deaths are from NCDs and probability of premature death from the four major NCDs is 21% [6]. Furthermore, NCDs explain more than half of the total mortality in 11 of the 54 African nations [5]. In sub-Saharan Africa, between 1990 and 2017, disability adjusted life years attributable to NCDs has increased by 67% [6].

NCDs have multiple genetic, environmental and behavioural determinants. Yet, the epidemiological shift observed in the last few decades is primarily attributable to changes in few major modifiable risk factors including dietary factors [1, 7]. A large body of evidence confirms that the rise in unhealthy diets is among the major drivers of the global NCD pandemic [7–10]. In 2017, 11 million adulthood deaths were attributable to dietary factors [7]. Inadequate intake of fruits, vegetables and dietary fibre and high intake of salt, sugar, alcohol and fats lead to NCDs [7–10]. Even within the domain of fatty acids, individual types have distinct physiological properties: high intakes of saturated (SFA) and trans (TFAs) fatty acids are strongly linked with dyslipidaemia while optimal intakes of polyunsaturated (PUFAs) and monounsaturated (MUFA) fatty acids have beneficial effects [8, 11, 12].

In 2003 WHO and Food and Agriculture Organization of the United Nations (FAO) proposed population intake goals for preventing NCDs [8]. According to the recommendation, optimal macronutrient intake ranges based on contribution to daily energy intake are 55–75% for carbohydrate, 15–30% for fat and 7–20% for protein. Calories from SFAs and TFAs have to be below 10 and 1% of the total intake, respectively and that of PUFAs should be within 6 to 10%. Further, energy derived from free sugars should not exceed 10%. Dietary cholesterol intake should be limited ($< 300$ mg/day) and adequate ($> 400$ g/day) intake of fruits and vegetables is recommended. Specific goals have been set for sodium chloride, total dietary fibre and ω-3 and ω-6 fatty acids as well [8].

In Africa where there is active rise in NCD-related mortality, limited information is available whether the existing food supply is in conformity with these goals or not. A recent study

that described the global food supply over the period of 1961 and 2013 reported that high income countries are moving towards more dietary diversification and reduced supply of sugars while low-income countries remain relatively unchanged or had moved towards poor diet combinations [13]. Furthermore, the ongoing rapid nutrition transition in LMIC has increased the double burden of malnutrition [14]. Accordingly, double-duty actions including optimizing intake of nutrients are required for combating multiple forms of malnutrition [15].

The current study evaluated the African food balances against population intake recommendations for macronutrients, free sugars, fatty acids, dietary cholesterol and fruits and vegetables defined for preventing diet-related NCDs, assessed trends over three decades (1990–2017), and provided regional (continental), sub-regional (geographic and gross national income classifications) and country-level estimates.

## Methods

### Study design

The analysis was made based on the food balance sheets (FBS) compiled by FAO for 45 of the 54 African countries for the period 1990 to 2017 [16, 17]. Data for the recent four years (2017–2020) had not been assembled and made available for public use and were not included in the analysis. Likewise, the data for Burundi, Comoros, Democratic Republic of Congo, Equatorial Guinea, Eritrea, Libyan, Seychelles, Somalia and Southern Sudan are not publicly available; accordingly, these countries are not represented in the study.

The per capita supply of 95 major food commodities (kg/capita/day) was downloaded for each country-year. Then the energy, carbohydrate, fat, protein, SFA, MUFA, PUFA and cholesterol contents were determined by referring to a standard food composition database [18]. The per capita supply of fruits and vegetables was also computed.

### Data source

Food Balance Sheet, also known as Food Disappearance data, is estimation of the food supply of a country in a given period. FAO based on multiple data sources including official reports, determines the production, import, export, changes in stocks and non-food uses for major food commodities and estimates the food available for human consumption in a territory. FAO annually publishes the supply statistics of more than 90 primary (e.g. eggs, milk) and processed commodities (e.g. butter) for about 180 countries as the average of supply over the past three-year period [19]. Food Balance Sheet is an important tool for appraising food situations at various levels (global, regional or national) and informing national food and agriculture policies.

### Estimation of nutrient composition

The supply of 95 commodities (kg/person/day) was converted into nutrient values (calorie, carbohydrate, protein, fat, SFA, MUFA, PUFA, sugar and cholesterol) using the US Department of Agriculture (USDA) Food Data Central food composition database [18]. The USDA database was preferred than other food composition databases (e.g. Infoods database of the FAO) because it provide comprehensive information on several food groups for all nutrients represented in the study. However, at times when a food item is missing from the database, the FAO food composition table for international use was used [19]. When a commodity aggregates two different foods (e.g. mutton and goat meat, oranges and mandarins, lemons and limes), the nutrient composition was estimated as the average of the two assuming equal weights. Similarly, when a group aggregates multiple food types (peas, beans, fresh water fish. . ..) the specific food types within that domain were identified from the FAO FBS handbook [19], and the mean

nutrient composition was determined assuming equal weights. Nutrient compositions for "others" (other cereals, other pulses, other oil crops, other meats etc. . .) were estimated by taking the arithmetic mean of the related commodities listed in the FBS.

The FBS provides all data on meat in terms of carcass weight that includes non-edible bones [19]. Carcass weight was converted into edible weight using the conversion factors recommended by USDA [20]. The conversion factor ranged from 70% for bovine meat to 100% for fish. For eggs conversion factor of 98.5 was used [20].

The amount of all energy-yielding macronutrients (carbohydrate, protein and fat) and alcohol available in the food supply for each country-year was converted into calories using Atwater specific factors [21]. Calories from various types of fatty acids (SFA, MUFA, PUFA) were also computed.

## Estimation of energy contribution of free sugars

WHO defines "free sugars" as "*all monosaccharides and disaccharides added to foods by the manufacturer, cook or consumer, plus the sugars that are naturally present in honey, syrups and fruit juices*" [8, 22]. In the current study, calories from free sugars were estimated by totalling the energy contents of honey and industrially produced sugars and sweeteners, with calories from monosaccharides and disaccharides contained in fruit juices. To the best of our knowledge, no information is available regarding what proportion of fruits are consumed as juices in Africa. Accordingly, based on the National Health and Nutrition Examination Survey (NHANES) of the US, we assumed that 35% of the total fruits were consumed as juice in all country-year [23].

## Supply of fruits and vegetables

The supply of fruits and vegetables (g/capita/day) was estimated by summing the balances of all specific fruits and vegetables represented in the FBS. In line with the approach used by WHO [8] and other similar studies [24, 25], starchy roots and tubers including cassava, yams, potato and sweet potato were not considered as vegetables.

## Data management and analysis

The food supply statistics for each country-year was downloaded from the old (1990–2012) [16] and new (2013–2017) [17] FAOSTAT databases as csv files and exported to SPSS v24 for analysis. On top of regional (continental) and country-level figures, sub-regional estimates were computed by grouping countries based on geographic location and national income levels. The UN sub-region classification was used to classify countries into Northern, Eastern, Southern, Western and Central sub-regions [26]. Furthermore, the World Bank's Gross National Income (GNI) classification for the year 2020 was employed for stratifying countries as low-, lower-middle-, upper middle- or high-income economies [27]. The sub-regional and economic classification of the countries is provided as a S1 File. Whenever estimates are provided by aggregating multiple countries, population-weighted analysis was used. Percentage change over the study period (1990–2017) was determined by dividing difference between the base- and end-year to the base value in 1990. PUFA to SFA ratio was computed by dividing the energy contribution of PUFA with that of SFA.

## Results

### Trends in energy supply

Between 1990 and 2017, the per capita supply of energy in Africa increased by 16.6% from 2,685 in 1990 to 3,132 kcal/person/day in 2017. The rates of increase were above the regional

average in Central (28.8%) and Western (22.2%) sub-regions. Figures from 2017 indicated that the energy supply (kcal/person/day) was highest in Southern (3,406) and lowest in Eastern (2,625) sub-regions. Over the period, Africa had also seen significant improvements in the supply of all energy-yielding nutrients. Protein supply increased by 19.0%, whereas carbohydrate and fat supplies rose by 16.7 and 14.6%, respectively.

Comparison based on national income levels indicated that in 2017 the energy supply (kcal/person/day) in low-income countries (2,771) was much lower than that of upper-middle- or high-income countries (3,448). However, rates of improvements for all energy yielding nutrients were substantially higher in low- than in high-income countries (Table 1).

## Calorie contribution of energy-yielding macronutrients

Between 1990 and 2017, the contribution of carbohydrates to the total daily energy supply in Africa remained constant around 73%. In 2017, the carbohydrate's contribution was relatively lower in upper-middle- or high-income (65.6%) and Southern Africa (66.1%) countries but both were within the acceptable range (55–75%) (Table 2).

Country-specific figures indicated, in 2017 contribution of carbohydrates exceeded the upper limit of 75%, in seven countries including Madagascar (83.8%), Rwanda (78.9%), Ghana (78.9%), Mozambique (77.4%) and Nigeria (76.8%) (S2 File).

At regional-level, the calorie contribution of protein was around 9% and the figure remained below the minimum target of 10% in almost all sub-regions and income levels, with the exception of the Northern Africa sub-region (10.1%).

At regional-level, the calorie contribution of fat also remained constant around 10%. However, in 2017, the figure exceeded the limit of 20% in upper-middle- or high-income countries and in Southern Africa sub-region. Country-level estimates indicated, 21 countries including Sao Tome and Principe (35.3%), Gambia (28.0%), Mauritius (26.5%) and Tunisia (25.7%) exceeded the limit of 20% (S2 File).

## Energy contribution of specific types of fatty acids

Table 3 shows the trends in per capita supply of SFA, MUFA and PUFA, expressed as contribution to total energy. In Africa, the calorie contribution of the fatty acids is found to be balanced. In 2017, 5.1% of the total calories came from UFA whereas, MUFA and PUFA contributed for 5.7 and 5.3%, respectively. Over the period (1990–2017) meaningful changes have not been seen at regional or sub-regional levels in terms of the energy derived from the three groups of fatty acids (Table 3).

In 2017, the energy from SFAs remained within the acceptable range (<10%) in all countries except in Sao Tome and Principe (26.0%). The supply of PUFAs, on the other hand was sub-optimal (< 6%) in 28 African countries including Madagascar (2.0%), Liberia (3.0%), Sierra Leone (3.0%), Ghana (3.0%) and Rwanda (3.1%) (S2 File).

The balance between SFA and PUFA in a diet can also be measured using PUFA to SFA ratio. Over the period, the ratio remained around 1:1 in Africa. In 2017, the ratio was relatively higher in upper-middle- or high-income countries (1.54:1) and in Southern (1.56:2) and Northern (1.56:1) sub-regions suggesting the dominance of PUFA. Conversely, the ratio was low (0.76:1) in the Western sub-region indicating the opposite.

## Energy contribution of free sugars

Over the period, the energy contribution of free sugars (%) in Africa remained constant around 7%. Throughout the years the figure exceeded the upper limit of 10% in upper-middle- or high-income countries and in Southern and Northern sub-regions. At country-level,

**Table 1. Trends in the supply of macronutrients in Africa, 1990–2017.**

| Supply of macronutrients | Year | | | | | | | % change |
|---|---|---|---|---|---|---|---|---|
| | 1990 | 1995 | 2000 | 2005 | 2010 | 2015 | 2017 | |
| Calorie supply (kcal/capita/day) | | | | | | | | |
| Africa | 2685 | 2791 | 2856 | 2942 | 3020 | 3144 | 3132 | 16.6 |
| National income level | | | | | | | | |
| Low-income | 2208 | 2233 | 2355 | 2462 | 2562 | 2765 | 2771 | 25.5 |
| Lower-middle-income | 2907 | 3063 | 3106 | 3194 | 3271 | 3338 | 3312 | 13.9 |
| Upper-middle- or high-income | 3025 | 3026 | 3118 | 3182 | 3237 | 3402 | 3448 | 14.0 |
| Sub-region | | | | | | | | |
| Northern | 3208 | 3283 | 3343 | 3412 | 3549 | 3713 | 3675 | 14.6 |
| Central | 2283 | 2320 | 2436 | 2628 | 2834 | 2940 | 2940 | 28.8 |
| Southern | 3004 | 2994 | 3087 | 3155 | 3201 | 3346 | 3406 | 13.4 |
| Eastern | 2267 | 2217 | 2286 | 2385 | 2436 | 2633 | 2625 | 15.8 |
| Western | 2700 | 3028 | 3102 | 3195 | 3261 | 3309 | 3300 | 22.2 |
| Carbohydrate supply (g/capita/day) | | | | | | | | |
| Africa | 495.0 | 518.3 | 527.4 | 536.6 | 549.3 | 579.0 | 577.6 | 16.7 |
| National income level | | | | | | | | |
| Low-income | 411.8 | 414.9 | 439.9 | 455.6 | 469.8 | 509.6 | 511.5 | 24.2 |
| Lower-middle-income | 538.6 | 575.0 | 576.7 | 585.8 | 600.0 | 620.9 | 616.7 | 14.5 |
| Upper-middle- or high-income | 513.4 | 510.0 | 526.0 | 518.4 | 518.5 | 562.8 | 574.4 | 11.9 |
| Sub-region | | | | | | | | |
| Northern | 603.0 | 612.4 | 619.7 | 638.5 | 656.3 | 691.8 | 684.2 | 13.5 |
| Central | 403.9 | 420.2 | 431.2 | 465.2 | 506.6 | 513.0 | 521.0 | 29.0 |
| Southern | 513.4 | 508.9 | 524.7 | 517.7 | 516.9 | 556.3 | 570.3 | 11.1 |
| Eastern | 431.0 | 424.8 | 440.7 | 450.0 | 453.5 | 491.1 | 488.2 | 13.3 |
| Western | 488.1 | 560.1 | 566.5 | 571.4 | 589.8 | 615.2 | 615.7 | 26.1 |
| Protein supply (g/capita/day) | | | | | | | | |
| Africa | 66.6 | 67.6 | 71.0 | 74.8 | 78.1 | 80.5 | 79.3 | 19.0 |
| National income level | | | | | | | | |
| Low-income | 54.9 | 55.6 | 59.0 | 62.7 | 67.0 | 70.8 | 71.4 | 30.0 |
| Lower-middle-income | 71.0 | 72.3 | 76.2 | 80.3 | 83.7 | 85.1 | 82.9 | 16.8 |
| Upper-middle- or high-income | 83.9 | 82.7 | 85.1 | 88.0 | 88.4 | 90.2 | 89.6 | 6.8 |
| Sub-region | | | | | | | | |
| Northern | 80.8 | 83.5 | 88.8 | 92.8 | 98.7 | 101.8 | 100.1 | 23.8 |
| Central | 57.6 | 55.3 | 62.2 | 66.5 | 72.9 | 83.5 | 80.3 | 39.4 |
| Southern | 82.9 | 81.2 | 83.6 | 86.8 | 86.9 | 88.0 | 87.9 | 6.0 |
| Eastern | 55.0 | 53.7 | 55.0 | 58.5 | 61.1 | 66.5 | 66.2 | 20.4 |
| Western | 65.1 | 68.6 | 73.1 | 77.8 | 81.1 | 79.2 | 77.9 | 19.7 |
| Fat supply (g/capita/day) | | | | | | | | |
| Africa | 55.3 | 56.8 | 58.9 | 63.1 | 64.7 | 63.7 | 63.4 | 14.6 |
| National income level | | | | | | | | |
| Low-income | 42.4 | 44.2 | 45.2 | 48.7 | 51.7 | 54.6 | 54.3 | 28.0 |
| Lower-middle-income | 60.2 | 61.2 | 64.2 | 68.7 | 69.4 | 66.3 | 65.9 | 9.5 |
| Upper-middle- or high-income | 73.6 | 77.4 | 79.9 | 88.9 | 94.7 | 91.6 | 92.1 | 25.2 |
| Sub-region | | | | | | | | |
| Northern | 65.3 | 68.0 | 70.3 | 68.3 | 73.6 | 74.8 | 74.8 | 14.6 |
| Central | 53.0 | 51.9 | 57.0 | 61.3 | 62.8 | 66.0 | 64.5 | 21.6 |
| Southern | 72.0 | 75.1 | 77.9 | 86.9 | 92.2 | 89.5 | 90.2 | 25.2 |

(*Continued*)

**Table 1.** (Continued)

| Supply of macronutrients | Year | | | | | | | % change |
|---|---|---|---|---|---|---|---|---|
| | 1990 | 1995 | 2000 | 2005 | 2010 | 2015 | 2017 | |
| Eastern | 41.5 | 39.4 | 39.8 | 45.4 | 48.4 | 50.9 | 51.4 | 23.8 |
| Western | 58.0 | 62.1 | 66.0 | 72.9 | 70.4 | 64.5 | 63.6 | 9.6 |

**Table 2. Contribution (%) of carbohydrate, protein and fat to total energy supply in Africa, 1990–2017.**

| Contribution to total calorie supply (%) | Year | | | | | | |
|---|---|---|---|---|---|---|---|
| | 1990 | 1995 | 2000 | 2005 | 2010 | 2015 | 2017 |
| Carbohydrate contribution (%) | | | | | | | |
| Africa | 72.9 | 73.6 | 73.2 | 72.2 | 72.0 | 73.0 | 73.2 |
| National income level | | | | | | | |
| Low income | 74.3 | 74.4 | 74.8 | 74.0 | 73.4 | 73.9 | 74.0 |
| Lower-middle income | 72.8 | 74.0 | 73.1 | 72.1 | 72.1 | 73.3 | 73.5 |
| Upper-middle- or high-income | 66.7 | 66.3 | 66.2 | 64.1 | 63.0 | 65.2 | 65.6 |
| Sub-region | | | | | | | |
| Northern | 72.4 | 71.8 | 71.3 | 72.0 | 71.2 | 71.9 | 71.9 |
| Central | 70.1 | 71.9 | 70.4 | 70.3 | 70.9 | 69.3 | 70.3 |
| Southern | 67.3 | 67.0 | 66.9 | 64.7 | 63.7 | 65.6 | 66.1 |
| Eastern | 75.3 | 76.3 | 76.7 | 75.0 | 74.0 | 74.3 | 74.1 |
| Western | 72.7 | 74.1 | 73.1 | 71.5 | 72.3 | 74.5 | 74.9 |
| Protein contribution (%) | | | | | | | |
| Africa | 8.9 | 8.6 | 8.8 | 9.0 | 9.2 | 9.1 | 8.9 |
| National income level | | | | | | | |
| Low income | 8.9 | 8.7 | 8.8 | 9.0 | 9.2 | 9.0 | 9.1 |
| Lower-middle-income | 8.8 | 8.4 | 8.7 | 8.9 | 9.1 | 9.0 | 8.8 |
| Upper-middle- or high-income | 9.8 | 9.6 | 9.6 | 9.8 | 9.8 | 9.5 | 9.3 |
| Sub-region | | | | | | | |
| Northern | 9.5 | 9.7 | 9.9 | 10.2 | 10.4 | 10.2 | 10.1 |
| Central | 9.2 | 8.3 | 8.9 | 8.9 | 9.0 | 10.2 | 9.7 |
| Southern | 9.7 | 9.5 | 9.5 | 9.7 | 9.6 | 9.4 | 9.2 |
| Eastern | 8.6 | 8.4 | 8.3 | 8.5 | 8.7 | 8.8 | 8.8 |
| Western | 8.4 | 7.9 | 8.3 | 8.6 | 8.7 | 8.3 | 8.1 |
| Total fat contribution (%) | | | | | | | |
| Africa | 17.9 | 17.6 | 17.8 | 18.5 | 18.6 | 17.6 | 17.5 |
| National income level | | | | | | | |
| Low-income | 16.7 | 16.7 | 16.3 | 16.9 | 17.3 | 16.9 | 16.7 |
| Lower-middle-income | 18.2 | 17.4 | 18.0 | 18.8 | 18.6 | 17.3 | 17.4 |
| Upper-middle- or high-income | 21.3 | 22.4 | 22.5 | 24.5 | 25.7 | 23.7 | 23.5 |
| Sub-region | | | | | | | |
| Northern | 18.0 | 18.5 | 18.8 | 17.8 | 18.4 | 17.9 | 18.0 |
| Central | 20.3 | 19.4 | 20.2 | 20.3 | 19.2 | 19.6 | 19.2 |
| Southern | 21.0 | 21.9 | 22.1 | 24.1 | 25.2 | 23.4 | 23.2 |
| Eastern | 15.9 | 15.2 | 14.9 | 16.2 | 17.1 | 16.5 | 16.7 |
| Western | 18.7 | 17.9 | 18.5 | 19.8 | 18.8 | 16.9 | 16.7 |

**Table 3. Calorie contribution (%) of different groups of fatty acids to the total energy supply in Africa, 1990–2017.**

| Contribution to total calorie (%) | Year | | | | | | |
|---|---|---|---|---|---|---|---|
| | 1990 | 1995 | 2000 | 2005 | 2010 | 2015 | 2017 |
| Saturated fatty acids contribution (%) | | | | | | | |
| Africa | 5.4 | 5.2 | 5.1 | 5.7 | 5.7 | 5.1 | 5.1 |
| National income level | | | | | | | |
| Low income | 4.9 | 4.9 | 4.6 | 5.0 | 5.2 | 4.8 | 4.8 |
| Lower-middle income | 5.6 | 5.3 | 5.4 | 6.0 | 5.9 | 5.2 | 5.3 |
| Upper-middle or high income | 5.5 | 5.4 | 5.5 | 6.1 | 6.3 | 5.8 | 5.6 |
| Sub-region | | | | | | | |
| Northern | 4.9 | 5.0 | 5.2 | 5.2 | 5.2 | 4.6 | 4.6 |
| Central | 6.1 | 5.6 | 5.6 | 5.6 | 5.3 | 5.2 | 5.1 |
| Southern | 5.4 | 5.3 | 5.4 | 6.0 | 6.2 | 5.7 | 5.5 |
| Eastern | 4.8 | 4.5 | 4.3 | 4.9 | 5.4 | 4.8 | 4.9 |
| Western | 6.2 | 6.0 | 5.8 | 6.6 | 6.2 | 5.6 | 5.5 |
| Monounsaturated fatty acids contribution (%) | | | | | | | |
| Africa | 5.6 | 5.5 | 5.6 | 5.8 | 5.9 | 5.7 | 5.7 |
| National income level | | | | | | | |
| Low-income | 5.4 | 5.4 | 5.3 | 5.4 | 5.5 | 5.6 | 5.5 |
| Lower-middle-income | 5.7 | 5.4 | 5.7 | 5.9 | 5.9 | 5.6 | 5.6 |
| Upper-middle- or high-income | 6.3 | 6.5 | 6.7 | 7.3 | 7.8 | 7.4 | 7.4 |
| Sub-region | | | | | | | |
| Northern | 5.5 | 5.7 | 5.9 | 5.3 | 5.6 | 5.7 | 5.8 |
| Central | 7.0 | 6.8 | 7.2 | 7.3 | 6.9 | 7.2 | 6.9 |
| Southern | 6.2 | 6.3 | 6.5 | 7.1 | 7.6 | 7.3 | 7.2 |
| Eastern | 4.9 | 4.5 | 4.5 | 4.9 | 5.2 | 5.1 | 5.1 |
| Western | 6.2 | 5.9 | 6.2 | 6.4 | 6.2 | 5.6 | 5.6 |
| Polyunsaturated fatty acids contribution (%) | | | | | | | |
| Africa | 5.3 | 5.3 | 5.4 | 5.5 | 5.4 | 5.2 | 5.3 |
| National income level | | | | | | | |
| Low income | 4.8 | 4.9 | 4.8 | 4.9 | 4.9 | 4.9 | 4.9 |
| Lower-middle income | 5.4 | 5.2 | 5.4 | 5.4 | 5.3 | 5.1 | 5.1 |
| Upper-middle or high income | 7.6 | 8.6 | 8.4 | 9.2 | 9.6 | 8.7 | 8.6 |
| Sub-region | | | | | | | |
| Northern | 6.3 | 6.4 | 6.2 | 5.7 | 6.0 | 6.1 | 6.2 |
| Central | 5.5 | 5.4 | 5.8 | 5.7 | 5.4 | 5.4 | 5.4 |
| Southern | 7.4 | 8.4 | 8.3 | 9.1 | 9.5 | 8.6 | 8.6 |
| Eastern | 4.7 | 4.6 | 4.6 | 4.9 | 4.9 | 5.1 | 5.1 |
| Western | 4.7 | 4.5 | 4.9 | 5.1 | 4.8 | 4.3 | 4.2 |
| Polyunsaturated to saturated fatty acid ratio | | | | | | | |
| Africa | 0.98 | 1.02 | 1.06 | 0.96 | 0.95 | 1.02 | 1.04 |
| National income level | | | | | | | |
| Low-income | 0.98 | 1.00 | 1.04 | 0.98 | 0.94 | 1.02 | 1.02 |
| Lower-middle-income | 0.96 | 0.98 | 1.00 | 0.90 | 0.90 | 0.98 | 0.96 |
| Upper-middle- or high-income | 1.38 | 1.59 | 1.53 | 1.51 | 1.52 | 1.50 | 1.54 |
| Sub-region | | | | | | | |
| Northern | 1.29 | 1.28 | 1.19 | 1.10 | 1.15 | 1.33 | 1.35 |
| Central | 0.90 | 0.96 | 1.04 | 1.02 | 1.02 | 1.04 | 1.06 |
| Southern | 1.37 | 1.58 | 1.54 | 1.52 | 1.53 | 1.51 | 1.56 |

(*Continued*)

**Table 3.** (Continued)

| Contribution to total calorie (%) | Year | | | | | | |
|---|---|---|---|---|---|---|---|
| | 1990 | 1995 | 2000 | 2005 | 2010 | 2015 | 2017 |
| Eastern | 0.98 | 1.02 | 1.07 | 1.00 | 0.91 | 1.06 | 1.04 |
| Western | 0.76 | 0.75 | 0.84 | 0.77 | 0.77 | 0.77 | 0.76 |

thirteen countries including Botswana (18.8%), Namibia (16.4%), Eswatini (15.2%) and South Africa (14.9%) exceed the limit (S3 File) (Table 4).

## Cholesterol supply

At regional-level, the total dietary cholesterol supply increased by 14% from 92.8 in 1990 to 105.7 mg/capita/day in 2017. The increase was also observed almost in all national income levels and sub-regions. Despite the progressive rise, cholesterol supply remained within the tolerable range set by WHO (<300 mg/day). Country-level estimates are given in a S4 File (Table 5).

**Table 4. Contribution of free sugars (%) to the total calorie supply in Africa, 1990–2017.**

| Contribution of free sugars to the total calorie (%) | Year | | | | | | |
|---|---|---|---|---|---|---|---|
| | 1990 | 1995 | 2000 | 2005 | 2010 | 2015 | 2017 |
| Africa | 6.9 | 6.8 | 7.0 | 7.0 | 7.1 | 7.0 | 7.1 |
| National income level | | | | | | | |
| Low-income | 5.2 | 5.2 | 5.2 | 5.9 | 5.8 | 6.0 | 6.0 |
| Lower-middle-income | 7.2 | 7.1 | 7.5 | 7.4 | 7.5 | 7.0 | 6.9 |
| Upper-middle- or high-income | 12.8 | 11.6 | 10.9 | 10.2 | 11.2 | 14.0 | 14.7 |
| Sub-region | | | | | | | |
| Northern | 10.2 | 9.5 | 10.3 | 10.6 | 10.8 | 10.5 | 10.5 |
| Central | 6.1 | 5.3 | 5.3 | 5.9 | 5.8 | 5.8 | 6.4 |
| Southern | 12.7 | 11.6 | 10.9 | 10.2 | 11.3 | 14.1 | 14.8 |
| Eastern | 5.6 | 5.9 | 6.3 | 6.1 | 6.2 | 6.2 | 6.2 |
| Western | 4.4 | 4.9 | 4.7 | 5.1 | 5.1 | 4.7 | 4.6 |

**Table 5. Dietary cholesterol supply (mg/capita/day) in Africa, 1990–2017.**

| Dietary cholesterol supply (mg/capita/day) | Year | | | | | | | % change |
|---|---|---|---|---|---|---|---|---|
| | 1990 | 1995 | 2000 | 2005 | 2010 | 2015 | 2017 | |
| Africa | 92.8 | 88.5 | 93.1 | 101.4 | 111.7 | 110.5 | 105.7 | 14.0 |
| National income level | | | | | | | | |
| Low income | 58.8 | 55.0 | 57.4 | 63.0 | 68.3 | 65.5 | 65.0 | 10.7 |
| Lower-middle income | 99.1 | 93.9 | 100.7 | 110.1 | 122.0 | 120.3 | 114.4 | 15.4 |
| Upper-middle or high income | 196.0 | 194.1 | 200.3 | 223.7 | 258.0 | 274.4 | 260.2 | 32.8 |
| Sub-region | | | | | | | | |
| Northern | 119.9 | 123.9 | 141.1 | 153.1 | 173.6 | 183.0 | 177.5 | 48.1 |
| Central | 91.1 | 74.8 | 83.6 | 87.8 | 101.8 | 127.8 | 116.6 | 28.0 |
| Southern | 187.6 | 184.4 | 189.8 | 213.8 | 246.1 | 263.7 | 250.5 | 33.5 |
| Eastern | 60.5 | 51.2 | 49.3 | 54.1 | 59.4 | 57.8 | 56.6 | -6.4 |
| Western | 82.2 | 78.8 | 82.5 | 92.6 | 99.5 | 87.1 | 83.3 | 1.4 |

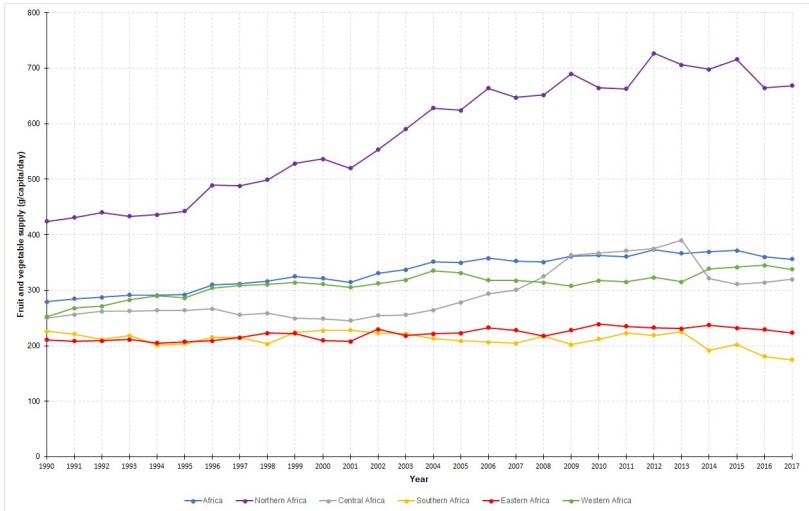

**Fig 1. Trends in the supply of fruits and vegetables in sub-regions of Africa, 1990–2017.**

## Fruits and vegetables supply

Over the period (1990–2017) the availability of fruits and vegetables per person in Africa rose by 27.5% from 279 to 356 g/capita/day. The trends in the supply of fruits and vegetables in different sub-regions and economic levels are shown in Figs 1 and 2. In terms of economic levels, the supply increased by 37.0 and 13.2% in lower-middle- and low-income countries, respectively, but had fallen by 20.8% in upper-middle- or high-income countries. Regarding geographical sub-regions, the supply substantially increased by 57.7% in Northern Africa. Considerable improvements have been seen in Western (33.3%) and Central (28.0%) sub-regions as well. Conversely, the improvements were modest (5.9%) in Eastern sub-region and even a considerable decline (22.6%) was seen in Southern sub-region. In general, the supply of

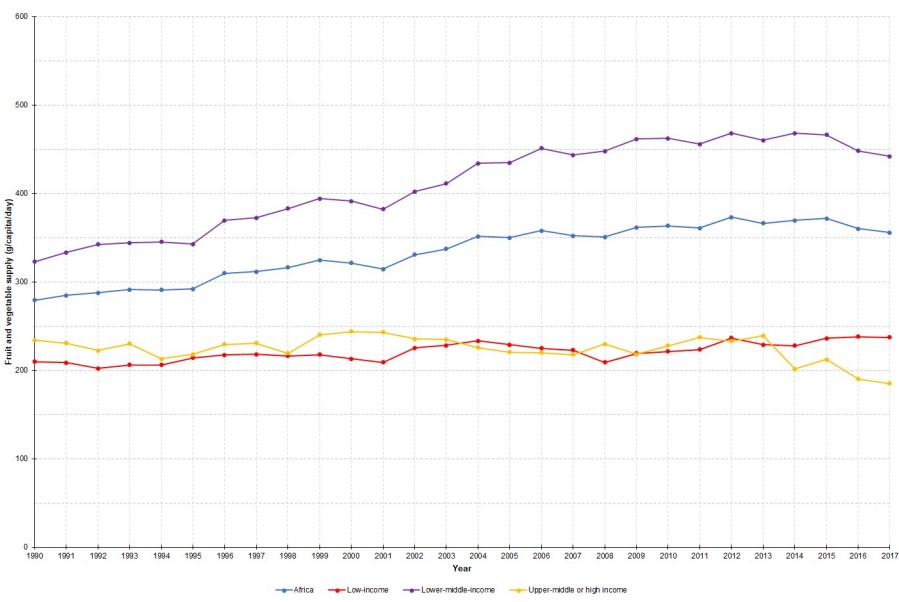

**Fig 2. Trends in the supply of fruits and vegetables across national income levels in Africa, 1990–2017.**

fruits and vegetables remained below the minimum target of 400 g/capita/day in all regions and economic levels except in Northern Africa and lower-middle-income countries. Country-level estimates are given as a S5 File.

## Discussion

The purpose of the current study was to evaluate the African food supply against population nutrient intake goals defined for preventing diet-related NCDs, and provide regional, sub-regional and country-level estimates.

The study indicated that between 1990 and 2017 the supply of all energy-yielding nutrients has considerably increased in the region, including in all national income levels. In low-income counties the actual supply remains lower but the rates of increase were higher than countries in higher income levels. The higher rates of increase can be explained by low baseline rates and recent economic growth being observed in many low-income African countries.

According to the recommendation of WHO, the fraction of energy derived from fat should not exceed 35% of the total intake and threshold of 20% is more compatible with good health [8]. This analysis suggested, between 1990 and 2017 the energy contribution of fat in Africa did not meaningfully change and in 2017 fat provided less than 35% of energy at all levels (regional, sub-regional or country). Yet, the figure exceeded 20% in 21 countries. Though the finding is potentially worrying, it is less alarming than what is being observed in the other regions of the world. According to an estimate, in 2013 Africa had the lowest (54.5 g/capita/day) fat supply in the world, and between 1961 and 2013 the supply was only increased by 15 g/capita/day. Conversely, fat supply markedly increased by 52, 48, 30 g/capita/day respectively, in South America, Asia and the Caribbean regions [28].

From the perspective of preventing both under- and over-nutrition, protein should contribute 10–15% of the daily energy intake [8]. However, the current analysis indicated that in 2017 energy derived from protein remained below 10% in all sub-regions except in Northern Africa. The finding indicates that the existing food supply in the region favours protein undernutrition more than overnutrition. A global estimate indicated, between 1961 and 2013 the protein supply per person in Africa was only increased by 31%, which was much lower than advances made in Asia (64%) and the Caribbean (46%) regions [29]. The supply in Africa (69.10 g/capita/day) was also the lowest among all other regions of the world [29]. An estimate based on FAO database also reported that globally the national meat supplies ranges from less than 10 kg/person/year in many low-income African countries, to more than 100 kg/person/year in high-income countries like USA and Australia. Similar patterns have also been seen for other protein rich animal source foods including milk, egg and fish [28]. The availability of protein rich foods is limited in many African countries due to multiple reasons including export of livestock to earn the much-needed foreign currency, low productivity in the dairy and poultry sectors and high price of animal source foods due to supply constraints [30].

Diets should contain an optimal amount of PUFAs, 6–10% of the total energy. PUFAs reduce plasma LDL and lessen the risk of cardiovascular diseases [8]. Apart from NCDs, ω-3 PUFAs are also required for brain development during early stages of life [31]. In Africa, with the exception of the Southern and Northern sub-regions, the PUFA supply appears to be sub-optimal (<6% of energy). According to our data, in Southern and Northern sub-regions, the PUFA supply was higher due to better access to fish and seed oils. Very few studies have so far evaluated the adequacy of PUFA supply at national or international levels [29, 32–34]. According to an estimate, globally only 1.3% of the total energy comes from PUFA, while MUFA and UFA, contribute 6.4% and 6.7%, respectively indicating suboptimal supply of PUFA is a global

concern [29]. The global analysis of PUFA intake found low intakes in many sub-Saharan Africa countries suggesting infrequent use of health cooking oils in the region [35].

Generally, it is assumed that the desirable ratio of PUFA to SFA in the diet is 1:1 [36, 37]. In the current study, the ratio was optimal at regional level but it was much lower in the Western sub-region (0.76:1) suggesting the dominance of SFAs. This can be due to relatively higher production of coconut oil in Western African countries including Côte D'ivoire, Nigeria and Gahanna. From NCDs perspective this is worrying because SFAs have strong tendency to increase LDL ("bad cholesterol") [8]. Analysis of national FBS data of Ethiopia [34] and Trinidad and Tobago [33] also indicated that the PUFA to SFA ratios were lower than one indicating the dominance of SFA in the respective national food supplies.

High dietary sugar intake causes positive energy balance and ultimately leads to NCDs through encouraging unhealthy weight gain [8]. It is recommended that the contribution of free sugars should not exceed 10% of the total energy [8] and further reduction to 5% provides additional benefits [22]. While free sugar intake in Africa remains within the acceptable range, the figure exceeded 10% in many countries, especially those in Southern and Northern sub-regions. This may partly explain the higher prevalence of NCDs in these sub-regions [5]. According to WHO, percentage of deaths from NCDs in Southern and Northern sub-regions is 70%, as compared to 41% in the entire region [5]. A study also indicated, between 1961 and 2007, calories derived from sugars had substantially increased in Northern Africa by 63% [38].

We estimated the energy contribution of free sugars by assuming that 35% of fruits were consumed as juice in all country-years. The assumption was made based on the finding of a national survey conducted in USA [23]. This decision might have made us to over or underestimate the intake of free sugars because the consumption pattern of whole fruits is likely to vary among different regions and countries. It can also change over time. Furthermore, as the consumption of sugars considerably varies across different stages of lifespan [39], the per capita supply statistics may not adequately capture the inter-individual variation in intake.

Even though the effect of dietary cholesterol on serum cholesterol has long been debated [40–42], the existing WHO's nutrient intake recommends for restricting intake below 300 mg/day [8]. According to our finding, between 1990 and 2017, the total dietary cholesterol supply in Africa has increased by 14% but it remained below the aforementioned threshold in all subregions and economic levels. Very few studies have so far explored the supply of dietary cholesterol at national-level [33, 35, 43]. In Trinidad and Tobago, between 1961 and 2007, the supply increased by 80% to 225 mg/day [33]; whereas the corresponding level in Finland was 440 mg/day [43]. According to a global estimate, the mean dietary cholesterol intake in 2010 was 228 mg/day and lowest intakes were in South Asian and East African nations [35].

Adequate intake of fruits and vegetables contributes to the reduction of energy density, promotes the consumption of dietary fibre, and reduces the risk of NCDs including obesity, cardiovascular diseases and possibly gastrointestinal cancers [8]. The current analysis suggested that, in Africa the fruits and vegetables supply is gradually improving but remains below the minimum target of 400 g/person/day in all sub-regions, except in the North. The better supply of fruits and vegetables observed in the North Africa is likely the reflection of the better economic status of the countries in the sub-region. A previous study estimated that in 2005 the fruit and vegetable availability was around 546 g/person/day globally, and the lowest supply was in sub-Saharan Africa (206 g/person/day) and South Asia (326 g/person/day). Conversely, the Middle East and North Africa had one of the highest supplies (735 g/person/day) [25]. It has also been projected that, by 2050 up to 1.9 billion people in sub-Saharan Africa could live in countries with inadequate availability of fruits and vegetables [25]. A multicounty study that enrolled 10 sub-Saharan Africa countries concluded that in all countries the per capita consumption of fruits and vegetables was below 50 g/day [31]. In addition to NCDs, low intake of

fruits and vegetables had been identified as a major driver of micronutrient deficiencies in Africa [44].

This study evaluated the African food supply against multiple population intake goals set for preventing NCDs. However, the following limitations should be taken into consideration while interpreting the findings. The overall analysis is made by considering food supply as a proxy indicator of food consumption; however, this assumption is not strictly true and might have caused overestimation of the intake of the nutrients. While FBS is a useful tool for international comparison and analysis over time, it does not take within country variation including geographic, seasonal and interpersonal differences, into account [19]. According to a review, multiple studies from Africa have already reported large subnational variation in the supply and consumption of foods and nutrients [45]. Furthermore, FBS does not take household-level food wastage and subsistent production of less pertinent food commodities, into consideration and this might have caused underestimation of intake.

In the current study, trends in dietary supply over the reference period were constructed by merging the old (1990–2012) [16] and new (2013–2017) [17] FAOSTAT databases. The two datasets have some methodological differences especially in relation to estimation of unbalanced amounts, source of population data and modelling of food stock and loss data [46]. However, in all of the trend analysis presented in this study, no abrupt interruption was observed between 2012 and 2013, suggesting that the use of the two datasets had no major effects on the timeseries analysis.

On top of the inherent limitations of FBS, we did not evaluate the African food supply against nutrient intake goals set for ω-3 and ω -6 fatty acids because a comprehensive food composition database is not available for these nutrients. Further, intake goals set for TFAs and fibres had not been evaluated because naturally occurring trans fats are less relevant to NCDs than their artificial counterparts [47]. Similarly, the intake of fibre is more affected by food processing rather than total food supply. While we compare the outcomes of interest across macroeconomic status of the countries, we used the GNI per capita categories for the year 2020. However, it is important to note that over the study period many countries have moved up in the income brackets and this might have affected the precision the time series comparisons made across GNI per capita categories. Irrespective of these methodological shortcomings, the study has provided comprehensive information on the African food supply from NCDs perspective and permitted for between-country and over time comparisons.

## Conclusion

Between 1990 and 2017, the per capita supply of calories substantially increased in Africa, including all sub-regions and economic categories. Most population intake goals set for preventing NCDs remained within acceptable range suggesting that many African countries are in the early stages of the nutrition transition. However, the supplies of fruits and vegetables and PUFAs are low and the increasing energy contributions of free sugars and fats are emerging concerns in specific sub-regions and countries of Africa.

## Supporting information

**S1 File. Sub-regional and economic classification of the 45 countries included in the study.**
(XLSX)

**S2 File. Country-specific estimates: Contribution (%) of carbohydrate, protein and fat to total energy supply in Africa, 1990–2017.**
(XLSX)

**S3 File. Country-specific estimates: Contribution of free sugars (%) to the total calorie supply in Africa, 1990–2017.**
(XLSX)

**S4 File. Country-specific estimates: Dietary cholesterol supply in Africa, 1990–2017.**
(XLSX)

**S5 File. Country-specific estimates: Fruit and vegetable supply in Africa, 1990–2017.**
(XLSX)

## Acknowledgments

The author acknowledges the Food and Agriculture Organization of the UN for making the food balance sheets data publicly available without restrictions.

## Author Contributions

**Conceptualization:** Samson Gebremedhin, Tilahun Bekele.

**Data curation:** Samson Gebremedhin, Tilahun Bekele.

**Formal analysis:** Samson Gebremedhin.

**Methodology:** Samson Gebremedhin, Tilahun Bekele.

**Software:** Samson Gebremedhin.

**Writing – original draft:** Samson Gebremedhin.

**Writing – review & editing:** Samson Gebremedhin, Tilahun Bekele.

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
