## [Decision Letter · Decision Letter 0]

9 Dec 2020

PONE-D-20-31495

Evaluating the African food supply against the nutrient intake goals set for preventing diet-related non-communicable diseases: 1990 to 2017 trend analysis

PLOS ONE

Dear Dr. Gebremedhin,

Thank you for submitting your manuscript to PLOS ONE. After careful consideration, we feel that it has merit but does not fully meet PLOS ONE’s publication criteria as it currently stands. Therefore, we invite you to submit a revised version of the manuscript that addresses the points raised during the review process.

We look forward to receiving your revised manuscript.

Kind regards,

Susan Horton

Academic Editor

PLOS ONE

Journal Requirements:

3. In the data analysis section, please clearly describe how the increases in nutrient supply are assessed against the regional recommendations. Currently, the statistical analysis only show percentage change which does not completely answer the primary research objective.

Reviewers' comments:

Reviewer's Responses to Questions

**Comments to the Author**

1. Is the manuscript technically sound, and do the data support the conclusions?

Reviewer #1: Yes

Reviewer #2: Yes

2. Has the statistical analysis been performed appropriately and rigorously? 

Reviewer #1: Yes

Reviewer #2: Yes

3. Have the authors made all data underlying the findings in their manuscript fully available?

Reviewer #1: Yes

Reviewer #2: Yes

4. Is the manuscript presented in an intelligible fashion and written in standard English?

Reviewer #1: Yes

Reviewer #2: Yes

5. Review Comments to the Author

Reviewer #1: Thank you for the opportunity to review this interesting paper. The authors have provided trends of energy and nutrient supply data relevant for NCDS in Africa. I think these findings could be useful to inform African countries and sub-regions as they transition to different income levels and potential diet shifts that may imply.

My comments are to help the authors improve upon the reporting and strength of the paper and to assist the editor in making a decision.

Two major comments: 1) Important trends not mentioned and discussed. From Tables 1, we see that the percent change in supply of all nutrients have been higher for low-income countries which decreases with increase in income levels except for fats where higher changes were seen in the two extremes of income classifications (low-income (28%) and upper-middle income (25.2%) categories). Similar country-income relevant trends are seen for the proportion of energy for specific nutrients. These specific trends are important for different income categories as they make various transitions. These need to be described in results and their implications discussed. Could authors speculate what may be driving the changes in food supply? Food trade or own production and what are the implications for sustainable diets?

2) The FAO adopted a new methodology for compiling the Food Balance Sheet data from 2014-2017 (http://www.fao.org/faostat/en/#data/FBS) . What are the implications of using data from the old and new methods in your results? Any associated limitations need to be included in the discussion section.

Minor comments:

1) Using FAO data (1961-2013), a recent study by Bentham et al, 2020 (https://www.nature.com/articles/s43016-019-0012-2) described the global food supply and identified important trends, including trends towards more diversification and reduced sugars of diets in high income countries while low income groups remain relatively unchanged or trends towards poor diet combinations. While a regional specific analysis is still important, the authors may want to discuss or mention how the regional trends differ/compare with the global and what the present trends mean for Africa or different income-level countries within Africa as they make further economic developments.

2) Table 1: Please check the total calorie supply for whole of Africa and the sub-region, Northern. These are exactly the same numbers.

3) Some minor typos that need checks. It would have been helpful to have line numbers to make reference very specific. Page 7, ‘…the amount of all all’… delete one “all”. Page 18, …’vegetables remained below below… delete one “below”. Abstract; conclusion: should be revised to read as “In Africa....”. If space allow, it will be useful to state that these are country level data and not individual consumption data or other form that notifies the reader.

4) References: Some revisions needed to ensure consistency especially where organizations are cited. #5, 8,11, 16, 19 need to include URL and access date to be consistent with 1, 13, 14, 15, 23, 24.

Thank you.

Reviewer #2: I find the rationale for conducting this study compelling and its results have a wide range of potential uses, including advocacy for governments of African nations to consider what policies are needed to encourage increased production and availability of foods that make healthy diets more available, accessible, affordable and desirable to their populations. I appreciate the occasional mention by the authors of the double burden of malnutrition and encourage them to reference the recent Lancet series on this topic. Both under-nutrition and over-nutrition contribute to the growing burden of diet-related NCDs in this continent (reference: Wells et al. 2020, The double burden of malnutrition: aetiological pathways and consequences for health, https://doi.org/10.1016/S0140-6736(19)32472-9) and this requires double-duty actions such as the policy changes recommended above. (reference: Hawkes et al. 2020, Double-duty actions: seizing programme and policy opportunities to address malnutrition in all its forms, https://doi.org/10.1016/S0140-6736(19)32506-1)

Methodology – Overall the authors appear to have conducted this study with careful consideration of the strengths and limitations of food balance sheet data from FAO. However, the decision by the authors to use an American population survey to estimate the proportion of fruits that are consumed as juice and apply this across all African countries and all 27 years is very difficult to justify, in my opinion. I recommend that the authors provide more data specific to Africa to show that this assumption is reasonable. The literature shows large changes over time in consumption of fruit juices globally, with an increase in countries like South Africa but still relatively low per capita consumption compared to the USA (for example, Fava Neves 2020 https://doi.org/10.1007/s40858-020-00378-1). Also, the very high relative cost of fruit juice and sugar-sweetened beverages in African countries, in contrast to North America (cf. Headey & Alderman, 2019, https://doi.org/10.1093/jn/nxz158) is another reason to believe that the authors’ assumption would over-estimate consumption of free sugars in many of these countries.

Discussion - The authors of this study do well in placing their findings in the context of other comparable studies, both for the Africa region as well as other regions. However, I recommend that they take these findings one step further in the discussion section and consider what are the contributing factors to the food supply issues observed for Africa region. What are the likely explanations for the results observed? The higher supply of fruits and vegetables in North Africa is remarkable and it would be good to describe some of the key reasons for why this sub-region has succeeded in increasing its supply.

In another example, supply constraints for protein-rich animal source foods contribute to the high costs of these foods, making them unaffordable for a large proportion of the population. What are some of the supply chain issues affecting these foods specifically? For example, fresh cow’s milk and eggs are highly perishable and low productivity in the dairy and poultry sectors of low-income countries contributes to the high prices of these foods. (Headey & Alderman 2019 https://doi.org/10.1093/jn/nxz158)

Including in the results or discussion section an analysis of what are the key food groups that are implicated in some of the sub-regional differences would further add to our understanding of what has contributed to these trends. For example, what foods are available in higher amounts in West African countries that contribute to the higher SFA amount compared to PUFA? I believe this is the higher production and availability of palm oil, but it would be helpful to the reader to confirm this, if possible.

I think it would be helpful to mention the fact that these are national estimates and do not adequately describe the large subnational variation in supply and consumption of these foods and nutrients. I recommend including a reference to studies from African populations that show the contrast in fat intake between wealthy and poor households or between adults in urban vs. rural areas. The article by Steyn & Mchiza (2014, https://doi.org/10.1111/nyas.12433) provides several examples of these contrasts at the subnational level.

Finally, given the current global context, I recommend that the authors also consider adding a couple of statements on the expected impact of the COVID-19 pandemic on food supplies in the Africa region.

Please see my detailed comments and suggested language edits in the attached file.

6. PLOS authors have the option to publish the peer review history of their article (what does this mean?). If published, this will include your full peer review and any attached files.

Reviewer #1: No

Reviewer #2: No

---

## [Author Response · Author response to Decision Letter 0]

19 Dec 2020

Point by point response to the comments of the reviewers

Reviewer #1: 

Comment 1: Important trends not mentioned and discussed. From Tables 1, we see that the percent change in supply of all nutrients have been higher for low-income countries which decreases with increase in income levels except for fats where higher changes were seen in the two extremes of income classifications (low-income (28%) and upper-middle income (25.2%) categories). Similar country-income relevant trends are seen for the proportion of energy for specific nutrients. These specific trends are important for different income categories as they make various transitions. These need to be described in results and their implications discussed. Could authors speculate what may be driving the changes in food supply? Food trade or own production and what are the implications for sustainable diets?

Response: The issue is now addressed in the Results (Page 10, past paragraph) and Discussion (Page 20, second paragraph) sections. 

Comment 2: The FAO adopted a new methodology for compiling the Food Balance Sheet data from 2014-2017 (http://www.fao.org/faostat/en/#data/FBS). What are the implications of using data from the old and new methods in your results? Any associated limitations need to be included in the discussion section.

Response: The limitation is now discussed in the Discussion section (Page 23, second paragraph)

Comment 3: Using FAO data (1961-2013), a recent study by Bentham et al, 2020 (https://www.nature.com/articles/s43016-019-0012-2) described the global food supply and identified important trends, including trends towards more diversification and reduced sugars of diets in high income countries while low income groups remain relatively unchanged or trends towards poor diet combinations. While a regional specific analysis is still important, the authors may want to discuss or mention how the regional trends differ/compare with the global and what the present trends mean for Africa or different income-level countries within Africa as they make further economic developments.

Response: thank you for recommending this key reference. Now it is cited in the introduction section (Page 5, third paragraph). 

Comment 4: Table 1: Please check the total calorie supply for whole of Africa and the sub-region, Northern. These are exactly the same numbers.

Response: We really sorry for this silly error. We have now corrected the values for the North Africa region (Table 1). 

Comment 5: Some minor typos that need checks. It would have been helpful to have line numbers to make reference very specific. Page 7, ‘…the amount of all all’… delete one “all”. Page 18, …’vegetables remained below below… delete one “below”. Abstract; conclusion: should be revised to read as “In Africa....”. If space allow, it will be useful to state that these are country level data and not individual consumption data or other form that notifies the reader.

Response: Thank you very much. All the typos are now corrected. 

Comment 6: References: Some revisions needed to ensure consistency especially where organizations are cited. #5, 8,11, 16, 19 need to include URL and access date to be consistent with 1, 13, 14, 15, 23, 24.

Response: The first group of references are books published by organizations while the second group are online resources. That’s why the approach of citation was different for the two groups. 

Reviewer #2: 

Comment 7: I appreciate the occasional mention by the authors of the double burden of malnutrition and encourage them to reference the recent Lancet series on this topic. Both under-nutrition and over-nutrition contribute to the growing burden of diet-related NCDs in this continent (reference: Wells et al. 2020, The double burden of malnutrition: aetiological pathways and consequences for health, https://doi.org/10.1016/S0140-6736(19)32472-9) and this requires double-duty actions such as the policy changes recommended above. (reference: Hawkes et al. 2020, Double-duty actions: seizing programme and policy opportunities to address malnutrition in all its forms, https://doi.org/10.1016/S0140-6736(19)32506-1)

Response: Thank you very much. The issue of double burden of diseases and the double-duty actions are now described in the introduction section (Page 5, third paragraph) and the two recommended articles are cited. 

Comment 8: Methodology – Overall the authors appear to have conducted this study with careful consideration of the strengths and limitations of food balance sheet data from FAO. However, the decision by the authors to use an American population survey to estimate the proportion of fruits that are consumed as juice and apply this across all African countries and all 27 years is very difficult to justify, in my opinion. I recommend that the authors provide more data specific to Africa to show that this assumption is reasonable. The literature shows large changes over time in consumption of fruit juices globally, with an increase in countries like South Africa but still relatively low per capita consumption compared to the USA (for example, Fava Neves 2020 https://doi.org/10.1007/s40858-020-00378-1). Also, the very high relative cost of fruit juice and sugar-sweetened beverages in African countries, in contrast to North America (cf. Headey & Alderman, 2019, https://doi.org/10.1093/jn/nxz158) is another reason to believe that the authors’ assumption would over-estimate consumption of free sugars in many of these countries.

Response: We agree that the use of American population survey to estimate the proportion of fruits that are consumed as juice and apply this across all African countries is difficult to justify. However, we had no other study from Africa or other comparable settings to estimate this key parameter. The only thing we could do is to remove the entire analysis on intake of free sugar from the manuscript or to keep it as it is and discuss the possible limitations of using the external US data. We opted for the latter because, even using the US data, comparison between countries, sub-regions, and income levels can would be somehow possible. However, we have already discussed (Page 21, last paragraph) the possible implication of using the US data for estimating the proportion of fruits that are consumed as juice. 

Comment 9: Discussion - The authors of this study do well in placing their findings in the context of other comparable studies, both for the Africa region as well as other regions. However, I recommend that they take these findings one step further in the discussion section and consider what are the contributing factors to the food supply issues observed for Africa region. What are the likely explanations for the results observed? The higher supply of fruits and vegetables in North Africa is remarkable and it would be good to describe some of the key reasons for why this sub-region has succeeded in increasing its supply.

Response: We have now provided further discussion to explain issues including high supply of fruits and vegetables in North Africa, low level of PUFA consumption in West Africa and high level of PUFA supply in North Africa. 

Comment 10: In another example, supply constraints for protein-rich animal source foods contribute to the high costs of these foods, making them unaffordable for a large proportion of the population. What are some of the supply chain issues affecting these foods specifically? For example, fresh cow’s milk and eggs are highly perishable and low productivity in the dairy and poultry sectors of low-income countries contributes to the high prices of these foods. (Headey & Alderman 2019 https://doi.org/10.1093/jn/nxz158)

Response: The issue is now discussed in the first paragraph of page 20, and the paper by Headey & Alderman 2019 is now cited. 

Comment 11: Including in the results or discussion section an analysis of what are the key food groups that are implicated in some of the sub-regional differences would further add to our understanding of what has contributed to these trends. For example, what foods are available in higher amounts in West African countries that contribute to the higher SFA amount compared to PUFA? I believe this is the higher production and availability of palm oil, but it would be helpful to the reader to confirm this, if possible.

Response: This is likely due to relatively higher production of coconut oil in western African countries including Côte D'ivoire, Nigeria and Gahanna. The same is now mentioned in the discussion section (Page 21, first paragraph). 

Comment 12: I think it would be helpful to mention the fact that these are national estimates and do not adequately describe the large subnational variation in supply and consumption of these foods and nutrients. I recommend including a reference to studies from African populations that show the contrast in fat intake between wealthy and poor households or between adults in urban vs. rural areas. The article by Steyn & Mchiza (2014, https://doi.org/10.1111/nyas.12433) provides several examples of these contrasts at the subnational level.

Response: This issue is now stated in the Discussion section (second paragraph, page 23)

Comment 13: Finally, given the current global context, I recommend that the authors also consider adding a couple of statements on the expected impact of the COVID-19 pandemic on food supplies in the Africa region.

Response: We fear discussing the impact of the COVID-19 pandemic on food supplies in Africa could take the paper out of context for two reasons: (1) The reference period that the study is focused (1990-2017) does not embrace the COVID-19 pandemic period. (2) limited scientific evidence is available to argue that the pandemic is negatively affecting the food supply in the continent. 

Comment 14: Please see my detailed comments and suggested language edits in the attached file.

Response Thank you very much for the inputs and commitment. We have accommodated all the language suggestions.

---

## [Editor Report · Decision Letter 1]

22 Dec 2020

PONE-D-20-31495R1

Evaluating the African food supply against the nutrient intake goals set for preventing diet-related non-communicable diseases: 1990 to 2017 trend analysis

PLOS ONE

Dear Dr. Gebremedhin,

Thank you for submitting your manuscript to PLOS ONE. After careful consideration, we feel that it has merit but does not fully meet PLOS ONE’s publication criteria as it currently stands. Therefore, we invite you to submit a revised version of the manuscript that addresses the points raised during the review process.

I have noted some minor typos and request you to fix these prior to accepting the manuscript.

We look forward to receiving your revised manuscript.

Kind regards,

Susan Horton

Academic Editor

PLOS ONE

Additional Editor Comments (if provided):

Thank you for responding to the reviewer comments very thoroughly. I noted 3 small typos to fix, and following resubmission of the manuscript with these small edits, the manuscript can be accepted for publication. The edits are as follows:

p21, line 2: I believe "lower" is correct, not "low"

p23, first sentence of the second full paragraph on the page: "trends in dietary supply over the reference period were constructed by merging...." not "was constructed"

p24: lines 1 and 2: I believe "fibre" is correct not "fibres"

---

## [Editor Report · Decision Letter 2]

26 Dec 2020

Evaluating the African food supply against the nutrient intake goals set for preventing diet-related non-communicable diseases: 1990 to 2017 trend analysis

PONE-D-20-31495R2

Dear Dr. Gebremedhin,

We’re pleased to inform you that your manuscript has been judged scientifically suitable for publication and will be formally accepted for publication once it meets all outstanding technical requirements.

Kind regards,

Susan Horton

Academic Editor

PLOS ONE

---

## [Editor Report · Acceptance letter]

2 Jan 2021

PONE-D-20-31495R2 

Evaluating the African food supply against the nutrient intake goals set for preventing diet-related non-communicable diseases: 1990 to 2017 trend analysis 

Dear Dr. Gebremedhin:

I'm pleased to inform you that your manuscript has been deemed suitable for publication in PLOS ONE. Congratulations! Your manuscript is now with our production department. 

Kind regards, 

on behalf of

Dr. Susan Horton 

Academic Editor

PLOS ONE